# Assessing the Impact of COVID-19 Public Health Stages on Paediatric Emergency Attendance

**DOI:** 10.3390/ijerph17186719

**Published:** 2020-09-15

**Authors:** Thérèse McDonnell, Emma Nicholson, Ciara Conlon, Michael Barrett, Fergal Cummins, Conor Hensey, Eilish McAuliffe

**Affiliations:** 1Centre for Interdisciplinary Research Education and Innovation in Health Systems, UCD School of Nursing, Midwifery & Health Systems, University College Dublin, D04 C7X2 Dublin, Ireland; emma.nicholson@ucd.ie (E.N.); ciara.conlon@ucd.ie (C.C.); Eilish.McAuliffe@ucd.ie (E.M.); 2Children’s Health Ireland at Crumlin, D12 N512 Dublin, Ireland; Michael.Barrett@olchc.ie; 3Women’s and Children’s Health, School of Medicine, University College Dublin, D04 C7X2 Dublin, Ireland; 4National Children’s Research Centre, D12 N512 Dublin, Ireland; 5REDSPOT (Retrieval, Emergency and Disaster Medicine Research and Development), Emergency Department, Limerick University Hospital, V94 F858 Limerick, Ireland; Fergal.Cummins@hse.ie; 6Children’s Health Ireland at Temple Street, D01 XD99 Dublin, Ireland; Conor.hensey@cuh.ie

**Keywords:** COVID-19, paediatric, public health, emergency medicine, delayed attendance, avoidance

## Abstract

This study outlines the impact of COVID-19 on paediatric emergency department (ED) utilisation and assesses the extent of healthcare avoidance during each stage of the public health response strategy. Records from five EDs and one urgent care centre in Ireland, representing approximately 48% of national annual public paediatric ED attendances, are analysed to determine changes in characteristics of attendance during the three month period following the first reported COVID-19 case in Ireland, with reference to specific national public health stages. ED attendance reduced by 27–62% across all categories of diagnosis in the Delay phase and remained significantly below prior year levels as the country began Phase One of Reopening, with an incident rate ratio (IRR) of 0.58. The decrease was predominantly attributable to reduced attendance for injury and viral/viral induced conditions resulting from changed living conditions imposed by the public health response. However, attendance for complex chronic conditions also reduced and had yet to return to pre-COVID levels as reopening began. Attendances referred by general practitioners (GPs) dropped by 13 percentage points in the Delay phase and remained at that level. While changes in living conditions explain much of the decrease in overall attendance and in GP referrals, reduced attendance for complex chronic conditions may indicate avoidance behaviour and continued surveillance is necessary.

## 1. Introduction

On 11th March 2020, a global pandemic was declared by the World Health Organisation (WHO) due to the outbreak of a novel coronavirus (SARS-CoV-2) or COVID-19. Despite public health appeals to reassure the public that it was safe to access health services, significant changes in emergency department (ED) utilization patterns have been observed [1,2,3,4,5].

Using national health authority administration data, two studies from Ireland and Portugal show a reduction in the proportion of all patients presenting at ED across all acuity presentations [1,6]. Factors contributing to lower utilization of health services include public health emergency directed population restrictive measures (e.g., halted travel/sports and closed schools/childcare facilities) [7]. Shifts in health-seeking behaviours also contribute, with an unwillingness to access healthcare driven by concerns about contagion in hospital settings [8,9]. Evidence from SARS and MERS epidemics have shown that risk preventative behaviours, such as hospital avoidance, were widely practised, with fear of infection associated with avoidance of healthcare facilities [10,11,12,13,14]. Of concern was an increase in cases triaged ‘very urgent/immediate’ among those admitted; however, without presenting complaint or diagnosis, this finding is difficult to interpret [1,6]. Delayed presentations and restricted access to health care has worrying implications (excess morbidity and mortality), particularly for children with serious or chronic conditions or protection concerns [9,15]. Delayed diagnosis and treatment for complex chronic conditions may result in rapid deterioration in the health of a child and can have fatal consequences [9]. To the best of our knowledge no study tracks the nature of attendance by children throughout the course of the public health response to the virus.

The aim of this paper is to assess how the nature of paediatric attendance at the ED altered throughout each of the COVID-19 related public health emergency stages, from the first confirmed case.

## 2. Materials and Methods

### 2.1. Data

A national database of paediatric emergency department (ED) attendance which would facilitate year on year detailed analysis was not available. With the assistance of on-site system managers, anonymised clinical and demographic data were extracted from the electronic patient records of all children aged under 16 attending three paediatric EDs and an urgent care centre (UCC) in Dublin (referred to as Dublin), and two mixed adult/paediatric EDs located in regional cities (Regional), Limerick and Cork. Approximately 28% of the Irish population live in Dublin, which has three dedicated public paediatric emergency departments and one paediatric UCC. County Cork has a population of approximately 540,000, representing 11% of the national population, while County Limerick has a population of approximately 200,000, representing 4% of the national population. The proportion of the population aged under 16 is comparable in the Cork and Limerick areas, and approximately 2% higher for the greater Dublin Area [16]. All three Dublin hospitals and the UCC are easily accessed by a large local population, both by car and public transport. However, the population served by the Regional hospitals is more dispersed, with many patients having a longer commute to the ED and more limited public transport options. This is reflected in the proportion of ED visits referred by general practitioners (GPs), with more patients attending a GP prior to visiting the Regional EDs. Medical card status provides us with some information on the socio-economic status of the areas served by each of the participating hospitals. Medical card ownership, which entitles access to health services at no charge and is granted primarily to those on low incomes [17], is above the national average in the areas served by the ED located in Limerick, though below the national average in areas served by the Cork ED. While the Dublin hospitals serve a diverse population and the mix of patients attending each hospital varies in terms of disadvantage, on average the population in areas served by these hospitals has a medical card ownership rate below the national average [16,17]. These five hospitals and the UCC represent approximately 48% of the national annual census of paediatric (under 16 years) ED attendances [18]. Appendix A illustrates the location of each participating site.

Data included date/time, gender, month and year of birth, linked episodes of care, mode of arrival, source of referral, presenting complaint, triage score, disposition (for all) and diagnosis (Dublin hospitals and UCC only). All EDs utilised a five-point triage scale (1 = life threatening to 5 = non-urgent).

The COVID-19 emergency public health stages (Figure 1) were directed by the Irish National Public Health Emergency Team (NPHET). The first COVID-19 case in Ireland was seen on 29th February 2020 and this analysis refers to the period in 2020 prior to this case as Pre-COVID. The Containment phase, the period from the first reported case to the closure of schools on 12th March, involved the identification and containment of all cases of COVID-19. The Delay phase focused on creating health system capacity through the cancellation of elective procedures and minimising the spread of the virus through social distancing, school and childcare facility closures, and advice to work from home where possible. Ireland introduced mandatory stay-at-home measures on 28th March, entering a seven-week Mitigation phase. One of the study hospitals in Dublin transferred out all its paediatric services to another location (including redirecting all paediatric ED patients to the two paediatric EDs and the UCC in Dublin) on this date to safely maximise capacity for adult COVID-19 patients at the co-located adult hospital. The country began the first stage of a five-stage reopening on 18th May.

### 2.2. Statistical Approach

The primary outcome was presented as mean daily attendance for each of the health stages from 1 January 2020 to 31 May 2020. Negative binomial regressions were estimated and adjusted for seasonal trends and year effects, and the incident rate ratio (IRR) was reported. Prior year comparison was adjusted for seasonality, with 2018 and 2019 attendance averaged to adjust for year specific variation. Secondary outcomes included referrals by a general practitioner (GP), admissions from the ED, presentations triaged as urgent and low urgency, and the level of attendance with ambulance as the mode of arrival. This analysis evaluated changes in the pattern of attendance in aggregate and also investigated regional variation in outcomes.

ED diagnosis data was available for the Dublin hospitals. However, as none of these hospitals used matching diagnostic coding, a detailed mapping exercise was conducted using International Classification of Disease (ICD) categorisation as a guide [19] and with input from ED and paediatric clinicians, to ensure consistent categorisation across hospitals (Appendix C). Categorisation was further consolidated into summarised categories of conditions, based on expected prevalence under the restrictive conditions imposed by the pandemic related public health responses. For example, all conditions predominantly contagious in nature were likely to be impacted in a similar manner by the low level of personal contact (“Infectious”), while the suspension of sporting and other physical activities, coupled with the requirement to remain close to home, were likely to have a similar effect on many types of injuries (“Trauma and Physical Activity”). Graphic analysis illustrates weekly attendance by summary category and variation on prior year by public health stage for the more detailed categories.

Stata16 (StataCorp, College Station, Texas, USA) was used to analyse the data and estimate the results. Due to sample size limitations, analysis of temporal trends in attendance by patients with chronic complex conditions was based on overall visits during each stage, with prior year comparison used to identify variation. Chronic complex condition (CCC) was defined as any medical condition that can be reasonably expected to last at least 12 months (unless death intervenes) and to involve either several different organ systems or one organ system severely enough to require specialty paediatric care, and probably some period of hospitalisation in a tertiary care centre [20]. Conditions categorized as CCC are detailed in Appendix C.

### 2.3. Ethics

Ethical approval has been granted by the COVID-19 National Research Ethics Committee, established by the Minister of Health in Ireland to deliver an expedited review process for COVID-19 research (reference code: 20-NREC-COV-034). Due to the collection of non-personal and anonymous data, explicit consent was not required for the data collected for this study. In accordance with data protection regulations, the data was anonymised at each site by relevant hospital staff before being transferred securely to the research team. 

## 3. Results

Paediatric ED attendance for the three-month period from March to May 2020 was 21,545, a drop of 46% on 39,772 for the same period in 2018/2019. Pre-COVID attendance (Jan/Feb 2020) was already below prior years (Table 1) with an IRR of 0.93 (CI 0.89–0.98), and below the comparable prior year period during the two-week Containment phase with an IRR also of 0.93 (CI 0.85–1.01). ED attendance dropped significantly below prior years during the Delay phase with an IRR of 0.58 (CI 0.54–0.64) during the two week period following the closure of schools and childcare facilities, and reached its lowest in late March, running 54% below prior years as the country entered the seven-week Mitigation phase or “lockdown”, with an IRR of 0.46 (CI 0.44–0.49). Both the number and proportion of attendances referred by a GP experienced a large and sustained drop during the Delay phase (IRR 0.37: CI 0.30–0.46), with 21% of visits referred, compared with 34% in prior years. Admissions were below prior years during the pre-COVID period (IRR 0.92: CI 0.86–0.97) and experienced a sustained reduction similar to that in attendances during the Delay phase, although the proportion of visits resulting in admission remained stable at 14%–15% throughout the public health stages. The proportion of attendances triaged as urgent also remained consistent and, with the exception of a brief increase during the Containment phase, the proportion of visits triaged as low urgency also remained stable throughout. The proportional attendance by children aged 13 to 15 dropped most substantially in the Delay and Mitigation stages; however, the age-profile of attendance during Phase One of Reopening was similar to pre-COVID proportions, albeit based on a much lower level of attendance.

The pattern of attendance at the Dublin (Table 2) and Regional hospitals (Table 3) throughout the public health stages was comparable, with two exceptions. Attendance at the Regional hospitals began to drop during the Containment phase (IRR 0.75: CI 0.67–0.83) while the number of visits at the Dublin hospitals remained similar to prior years, and the increase in attendance during Phase One of Reopening at the Regional hospitals was also below that of the Dublin hospitals (Regional IRR 0.5: CI 0.44–0.56; Dublin IRR 0.6: CI 0.55–0.66). While the proportion of visits referred by a GP at the Dublin hospitals was below that of the Regional hospitals for all periods, the level and proportion of referrals began to drop at the Regional hospitals during the Containment phase (IRR 0.77; CI 0.66–0.91). Both Dublin and the Regional hospitals experienced a significant drop in GP referrals during the Delay phase, and the proportion of GP referred visits remained 9 and 11 percentage points respectively below the prior year rates during Phase One of Reopening. The rate of admissions from the ED at the Regional hospitals was above prior years throughout all public health stages, while the proportion of attendances resulting in admission at the Dublin hospitals was similar to prior years.

Figure 2 charts the percentage change compared to prior years for each of the main categories of diagnosis for the Dublin hospitals by public health stage, with a detailed breakdown reported in Appendix B. For patients with multiple diagnoses, the first diagnosis was used in this analysis. Figure 3 charts weekly attendance at a more summarised category of diagnosis throughout the public health stages from January to June 2020, with conditions grouped based on their expected prevalence within the context of the restrictive living conditions.

Attendance for all categories of diagnosis decreased during the Delay phase by between 27–62%. Virus related attendances typically represented between 33–47% of overall ED attendance during this five-month period in prior years. Viral related attendances form the majority of the summary category Infectious, which saw the greatest drop in the Delay phase, and this explains much of the overall decrease in attendances. Presentations within this summary category continued at a low level, as families stayed at home and schools and workplaces remained closed. ED admissions dropped for these conditions, though the proportion of Infectious admissions did increase. Sepsis presentations were comparable to prior periods. Attendance related to Trauma/Physical Activity also dropped during the Delay phase but began to rise steadily during the Mitigation phase. ED attendances relating to Personal Safety, such as poisonings, overdoses, and insertion/ingestion of foreign objects, accounting for approximately 2% of visits, dropped during the Mitigation phase, and rose somewhat during Phase One of Reopening. Mental health presentations, mainly by children aged 13−15, typically accounting for less than 1% of ED visits (three per day), dropped to a mean of just one per day during the Delay phase, rose slightly during the Mitigation phase but remained below pre-COVID and prior year levels in late May. The proportion of mental health presentations (13−15 years) admitted during the Delay and Mitigation phases was higher at 49%, compared with 42% in prior years.

The summary category Medical, which groups clinical presentations that may be unaffected by the environmental changes imposed by the public health measures, dropped by 43% during the Delay phase, increased week on week during the Mitigation phase, but remained 21% below prior year in Phase One of Reopening. Medical ED visits typically account for 17% of overall attendance and, while the level of attendance in prior years was stable from September to May, visits have dropped over the summer months to a level comparable to that of Phase One of Reopening. The rate of admission for these conditions was 27% during the Delay phase, compared with prior year of 23%. Admission rates returned to prior year levels during the Mitigation phase.

As patients with chronic complex conditions account for less than 0.5% of overall paediatric attendances, statistical analysis is challenging due to the small number of visits within this category. Table 4 therefore presented total visits per public health stage compared with the average for the corresponding periods in 2018/19. Pre-COVID ED visits for these conditions were 25% below prior years (Table 4). Attendance dropped during the Delay stage and remained down on prior years during the Mitigation phase and Phase One of Reopening, with the period March to May 2020 running 45% below prior years. Most of this decrease related to endocrine and oncology presentations, with 89% of patients attending during this period admitted (78% in 2018/2019) and 78% triaged as urgent (74% in 2018/19).

## 4. Discussion

This study analyses trends in paediatric ED attendance from before the first confirmed case of COVID-19 in Ireland and throughout the stages of emergency public health restrictions in the three-month period that followed. The greatest reduction occurred during the Delay phase, when the Irish authorities were concerned that hospitals would be overwhelmed, and public health messaging was designed to create capacity within the health system (Figure 1). Furthermore, as availability of transport to and from EDs is a concern for many parents [21,22], public health restrictions on travel and fear of using public transport may have led to some parents deciding not to access needed healthcare [23]. The decrease in attendances for common childhood complaints, such as respiratory and digestive illness and trauma and physical activity related injuries, may be largely explained by changes in behaviour due to restrictions on movement as well as closure of schools, creches, and cultural, sporting and social activities where such illnesses may typically have an opportunity to spread. Influenza surveillance reports for this period indicate a very low incidence for all age groups, particularly children aged under 14 [24]. The overall acuity of presentations and rate of hospital admission remained relatively stable throughout each public health stage.

Although the incidence of COVID-19 among children is low, with 2% of confirmed cases in Ireland as of 5th June aged under 15 [25], public health advice issued during the pandemic, particularly during the Delay phase, did not differentiate between adults and children and urged patients not to attend health services in person if they displayed symptoms associated with respiratory illness. Hospital avoidance due to fear of infection has been documented in previous epidemics [13], with patients also voicing concern about not wanting to burden the health system. However, such avoidance is problematic if it places a child at risk, and identifying these children is challenging as the number of patients in this category is small. If the aim of the public health messaging was to deter parents from bringing their children to the ED, the sharp drop in the Delay phase does suggest that avoidance occurred. Nonetheless, it is questionable whether this was an appropriate aim for paediatric services, with the drop in attendance for endocrine, neurology, haematology and oncology and the increase in the rate of admissions for complex chronic conditions, in particular, of concern. However, enhanced speciality support (unburdened of scheduled care commitments), in the absence of an established paediatric complex care programme, may explain some of the decrease in attendance. Further longitudinal research is needed to monitor this trend over the duration of the pandemic to determine any changes in morbidity and mortality outcomes.

The reduction in mental health presentations and the increase in the proportion of those admitted requires attention. Children and adolescents may be particularly vulnerable to the impact of restrictions on social contacts, closure of schools, and isolation on their mental health [26], the extent of which is still not known, and therefore monitoring these presentations throughout the duration of the pandemic is critical. The level of mental health presentations during the study period is comparable to the summer/early autumn period in prior years and may reflect the incidence of mental health presentations during school holidays. However, as the admission rate of these presentations is higher than prior years, potentially indicating a greater severity of presentation, this may be reflective of delayed presentation. The lack of community mental health services and support due to the constraints of the public health measures may also explain this increased severity. Increased admissions may, however, indicate greater capacity to admit, particularly cases that may be perceived as “borderline admit”.

The decline in the proportion of GP referred visits may be due to an overall decrease in attendance by paediatric patients at general practice. Many GPs moved to remote consultations from the onset of the pandemic, conducting fewer face-to-face consultations. GPs also played a critical role in referring probable cases of COVID-19 for testing, so accessibility for non-COVID illness may have been an issue. However, the decreased GP referral rate may also be somewhat attributable to a change in the referral process. GP referrals are captured on ED systems on the presentation of a referral letter and, as a GP referral entitles patients to a waiver of the usual ED charge of €100 for those without a medical card, capture of such referrals on ED systems is generally robust. However, some remote consultations may have resulted in advice to attend the ED without a GP issuing an accompanying letter.

### Implications for Policy and Practice

This study analysed almost half of the national paediatric ED attendances, and therefore provides robust confirmation of the decrease in ED utilisation during each phase of the COVID-19 stages. The findings are relevant for planning for any future spikes in COVID-19 as well as possible future global pandemics, highlighting the need for careful consideration of the possible impact of public health messaging on children’s health.

The change in the rate of referrals from GPs may reflect the diversion of general practice resources to managing the response to COVID-19, and their accessibility. Future health service planning should prioritise general practice accessibility. The drop in attendance for complex chronic conditions may be difficult to address, and policy makers will need to ensure public messaging does not deter parents with children displaying symptoms indicative of such conditions from seeking potentially life-saving diagnosis and treatment. Beyond the pandemic there may be an opportunity to learn from the experience of the past months and to encourage more efficient use of the ED by parents. Prior to the onset of COVID-19, attendances at EDs had been increasing, with some public health interventions and changes in service provision aimed at reducing this rise [27,28]. Understanding the changes in parents’ decision-making and behaviour, including where they sought help and advice in place of the ED, may provide useful learning for supporting care for childhood illness outside of the ED setting.

## 5. Limitations

Data relating to diagnostic categories was only available for the Dublin hospitals and therefore, these findings may not reflect regional outcomes. However, as the largest paediatric health service providers in the country, these hospitals and the urgent care centre account for a sizeable portion of ED attendances nationally. Additionally, while all efforts were made to account for reporting errors in this data, a small number of coding errors may remain. The small sample size of visits for chronic complex conditions presents a challenge, though monitoring such attendances is critical given the potential impact of delayed attendance on the health of these children.

## 6. Conclusions

Presentations to the ED have declined sharply, particularly among children, in the weeks immediately following the onset of the COVID-19 pandemic. Delayed or reduced access to paediatric emergency care can have severe consequences and concern has been expressed that the reduction in ED attendance may be due to avoidance behaviour. This study uses a unique sample of paediatric ED attendance to understand changes in ED presentations from the most restrictive phase of public health COVID-19 response through to the beginning of reopening. While a reduction in attendance for respiratory conditions, infectious diseases and certain injuries can be attributed to the restrictive living conditions imposed in response to COVID-19, a sustained decrease in medical attendance, and for complex chronic conditions in particular, may be indicative of avoidance behaviour. Decreased referrals from GPs may be indicative of reduced accessibility to general practice due to the pressure on GPs to manage the response to COVID-19. Surveillance of the nature of paediatric ED attendance must continue to ensure the identification of those at risk due to delayed attendance and to inform public health messaging as the burden placed on families of living through a pandemic persists.

## 7. Future Research

There is little evidence of the longer-term impact on health seeking behaviour during public health emergencies, and changes in attendance patterns, admission rates and mortality need to be monitored as the pandemic progresses over the coming months. This analysis forms part of a 12-month study [29] which will monitor paediatric ED attendance as the prevalence of and living conditions associated with COVID-19 evolve. Additionally, there is a need to understand any behavioural changes from the perspective of parents and families to provide context for changes in attendance patterns. Parental preferences when accessing unscheduled healthcare are being assessed as part of a related study [30] and a survey of parents in order to understand health seeking behaviour by parents for their children during COVID-19 forms part of this current study [23,29].

The decline in paediatric ED attendance is clear and the restrictions on movements and social gatherings, coupled with normal seasonal patterns of attendance, explains much of this variance. General practice referrals to ED have significantly decreased. This large nationally representative study does indicate that attendance by the less frequently encountered chronic complex patients may still be of concern during the pandemic. However, the potential for the systemic changes and the COVID-19 public health response to have a long term detrimental effect on children’s and young people’s health may still be underappreciated [31] and continued surveillance of paediatric ED attendance is needed, particularly as the risk of a further wave of COVID-19 increases.

## Figures and Tables

**Figure 1 ijerph-17-06719-f001:**
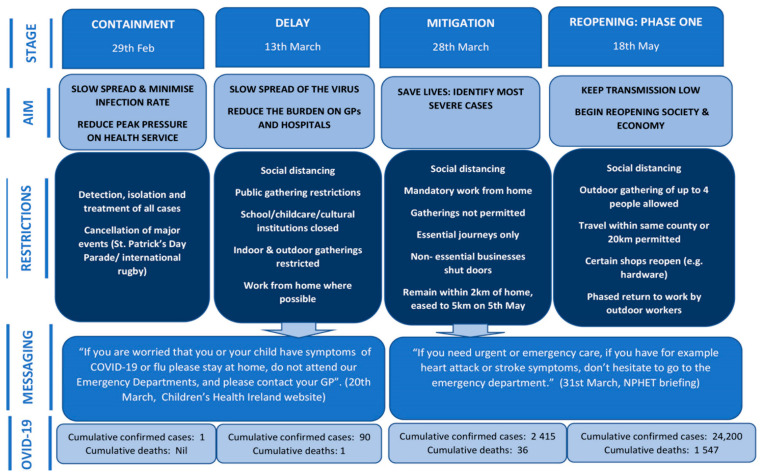
Public health stages of COVID-19 response.

**Figure 2 ijerph-17-06719-f002:**
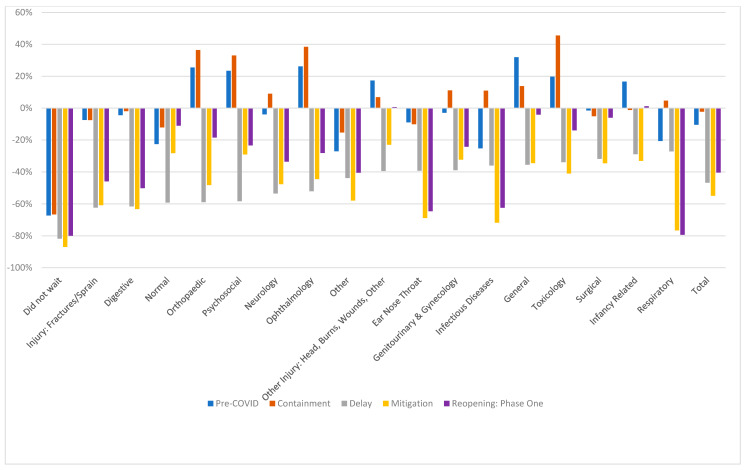
Change by Diagnosis from prior year by Public Health Stage. Prior year 2018/2019 is average daily attendance for comparable period in 2018 and 2019.

**Figure 3 ijerph-17-06719-f003:**
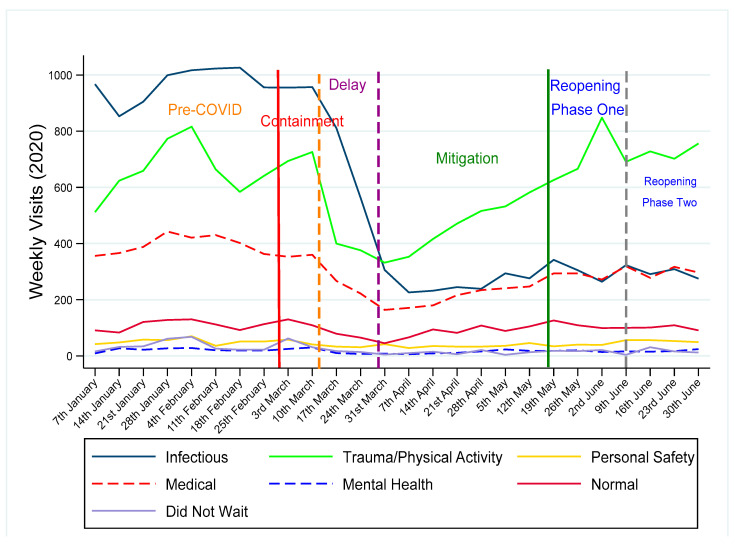
Emergency department attendance by category of diagnosis. Infectious includes respiratory, ENT, infectious diseases and digestive; Trauma/Physical Activity refers to sprains, fractures and bruising, head injuries, burns, lacerations and orthopaedic; Personal Safety includes toxicology, foreign objects (insertion/ingestion) and child protection/assault; Medical includes dermatology, genitourinary & gynaecology, neurology, surgical, general, ophthalmology and infancy related; Mental Health refers to psychosocial presentations; Did Not Wait refers to patients who registered on arrival but did not wait to see a doctor. This varies with the average wait time and is generally higher during busy period.

**Table 1 ijerph-17-06719-t001:** Characteristics of average daily paediatric emergency department attendance by public health COVID-19 response stage.

	Pre-COVID			Containment			Delay			Mitigation			Reopening: Phase One		
Date from	01/01/2020			29/02/2020			13/03/2020			28/03/2020			18/05/2020		
Date to	28/02/2020			12/03/2020			27/03/2020			17/05/2020			31/05/2020		
Number of Days	59			13			15			51			14		
**Daily Visits**			*%*			*%*			*%*			*%*			*%*
**Observed Mean**	396		100%	370		100%	232		100%	195		100%	257		100%
**Prior Years Mean**	444		100%	396		100%	435		100%	435		100%	450		100%
**IRR**	0.93			0.93			0.58			0.46			0.58		
**Confidence Interval**	0.89	0.98		0.85	1.01		0.54	0.64		0.44	0.49		0.53	0.63	
***p*-value**	0.004			0.086			0.000			0.000			0.000		
**GP Referrals**															
**Observed Mean**	133		34%	115		31%	49		21%	42		21%	54		21%
**Prior Years Mean**	157		35%	131		31%	149		34%	140		32%	143		32%
**IRR**	0.88			0.85			0.37			0.30			0.38		
**Confidence Interval**	0.78	0.99		0.69	1.07		0.30	0.46		0.26	0.34		0.31	0.47	
***p*-value**	0.033			0.163			0.000			0.000			0.000		
**Admitted**															
**Observed Mean**	57		14%	54		15%	34		15%	29		15%	36		14%
**Prior Years Mean**	63		14%	60		14%	64		15%	62		14%	64		14%
**IRR**	0.92			0.89			0.55			0.46			0.57		
**Confidence Interval**	0.86	0.97		0.79	0.99		0.49	0.62		0.43	0.49		0.51	0.64	
***p*-value**	0.004			0.030			0.000			0.000			0.000		
**Triage urgent (score 1 or 2)**															
**Observed Mean**	85		21%	78		21%	47		20%	38		19%	53		20%
**Prior Years Mean**	91		20%	80		19%	88		20%	87		20%	88		19%
**IRR**	0.97			0.95			0.58			0.44			0.60		
**Confidence Interval**	0.91	1.03		0.85	1.06		0.52	0.65		0.41	0.48		0.54	0.68	
***p*-value**	0.255			0.340			0.000			0.000			0.000		
**Triage low urgency (score 4 or 5)**															
**Observed Mean**	150		38%	146		40%	90		39%	77		39%	105		41%
**Prior Years Mean**	164		37%	150		35%	164		38%	173		40%	184		41%
**IRR**	0.97			0.98			0.61			0.46			0.58		
**Confidence Interval**	0.91	1.03		0.88	1.10		0.54	0.68		0.43	0.50		0.52	0.65	
***p*-value**	0.328			0.781			0.000			0.000			0.000		
**Arrival by ambulance**															
**Observed Mean**	24		6%	26		7%	18		8%	12		6%	14		5%
**Prior Years Mean**	25		6%	24		5%	25		6%	23		5%	26		6%
**IRR**	1.13			1.25			0.87			0.58			0.63		
**Confidence Interval**	1.05	1.22		1.10	1.42		0.76	1.00		0.53	0.64		0.53	0.73	
***p*-value**	0.002			0.001			0.057			0.000			0.000		
**Other statistics**	*n*	Std Dev		*n*	Std Dev		*n*	Std Dev		*n*	Std Dev		*n*	Std Dev	
**ICU admissions in period (actual)**	34			11			10			20			4		
**ICU admissions per day (mean)**	0.6	0.6		0.8	0.8		0.7	0.6		0.4	0.6		0.3	0.5	
**Age category (% proportion)**															
**Under 2**	107	16	27%	93	13	25%	76	13	33%	55	9	28%	63	11	25%
**Age 2 to 5**	86	12	22%	81	9	22%	59	13	25%	45	9	23%	50	6	19%
**Age 5 to 12**	130	29	33%	122	24	33%	71	14	31%	66	15	34%	95	17	37%
**Age 13 to 15**	74	21	19%	74	22	20%	27	8	12%	29	9	15%	49	8	19%

*n* is daily unless otherwise stated. 95% confidence intervals; prior year 2018/2019 are averaged mean attendance; percentages are of the total number of visits for the period; while Phase One is a three-week period from 18th May to 8th June, this analysis covers the period to 31 May 2020; estimation is by negative binominal regression and IRR is the incident rate ratio; admissions, including ICU admissions, are from emergency department (ED) only.

**Table 2 ijerph-17-06719-t002:** Paediatric emergency department attendance at Dublin hospitals by public health COVID-19 response stage.

	Pre-COVID			Containment			Delay			Mitigation			Reopening: Phase One		
Date from	01/01/2020			29/02/2020			13/03/2020			28/03/2020			18/05/2020		
Date to	28/02/2020			12/03/2020			27/03/2020			17/05/2020			31/05/2020		
Number of Days	59			13			15			51			14		
**Daily Visits**			%			%			%			%			%
**Observed Mean**	321		100%	310		100%	186		100%	158		100%	216		100%
**Prior Years Mean**	359		100%	317		100%	351		100%	351		100%	362		100%
**IRR**	0.94			0.97			0.59			0.47			0.60		
**Confidence Interval**	0.89	0.99		0.88	1.06		0.54	0.64		0.44	0.49		0.55	0.66	
***p*-value**	0.013			0.522			0.000			0.000			0.000		
**GP referral**															
**Observed Mean**	89		28%	80		26%	32		17%	25		16%	36		17%
**Prior Years Mean**	105		29%	88		25%	100		28%	95		27%	95		26%
**IRR**	0.88			0.89			0.36			0.28			0.37		
**Confidence Interval**	0.76	1.02		0.68	1.17		0.27	0.47		0.23	0.32		0.28	0.49	
***p*-value**	0.097			0.417			0.000			0.000			0.000		
**Admitted**															
**Observed Mean**	37		12%	37		12%	21		11%	18		11%	23		11%
**Prior Years Mean**	43		12%	42		12%	43		12%	42		12%	43		12%
**IRR**	0.86			0.88			0.50			0.42			0.52		
**Confidence Interval**	0.80	0.92		0.78	1.00		0.43	0.57		0.38	0.45		0.45	0.59	
***p*-value**	0.000			0.045			0.000			0.000			0.000		
**Triage urgent (score 1 or 2)**															
**Observed Mean**	64		20%	60		19%	37		20%	30		19%	43		20%
**Prior Years Mean**	70		19%	61		18%	68		19%	66		19%	67		18%
**IRR**	0.95			0.95			0.59			0.46			0.64		
**Confidence Interval**	0.89	1.02		0.84	1.07		0.52	0.67		0.43	0.50		0.56	0.50	
***p*-value**	0.142			0.413			0.000			0.000			0.000		
**Triage low urgency (score 4 or 5)**															
**Observed Mean**	141		44%	140		45%	83		44%	71		45%	99		46%
**Prior Years Mean**	153		43%	139		41%	155		44%	162		46%	171		47%
**IRR**	0.97			1.00			0.59			0.45			0.59		
**Confidence Interval**	0.91	1.03		0.89	1.13		0.53	0.67		0.42	0.49		0.52	0.66	
***p*-value**	0.320			0.939			0.000			0.000			0.000		
**Arrival by ambulance**															
**Observed Mean**	18		6%	19		6%	13		7%	9		6%	12		5%
**Prior Years Mean**	18		5%	16		5%	17		5%	17		5%	18		5%
**IRR**	1.20			1.37			0.94			0.66			0.74		
**Confidence Interval**	1.10	1.30		1.18	1.59		0.80	1.11		0.60	0.74		0.62	0.88	
***p*-value**	0.000			0.000			0.478			0.000			0.001		
**Age category (% proportion)**		Std Dev			Std Dev			Std Dev			Std Dev			Std Dev	
**Under 2**	86	13	27%	78	12	25%	61	12	33%	44	9	28%	52	10	24%
**Age 2 to 5**	69	10	22%	67	7	22%	47	11	25%	36	8	23%	42	5	19%
**Age 5 to 12**	107	25	33%	102	21	33%	58	14	31%	54	14	34%	83	16	38%
**Age 13 to 15**	60	18	19%	63	17	20%	22	7	12%	22	9	15%	40	7	18%

*n* is daily unless otherwise stated. 95% confidence intervals; prior year 2018/2019 are averaged mean attendance; percentages are of the total number of visits for the period; while Phase One is a three-week period from 18th May to 8th June, this analysis covers the period to 31 May 2020; estimation is by negative binominal regression and IRR is the incident rate ratio; admissions are from ED only.

**Table 3 ijerph-17-06719-t003:** Paediatric emergency department attendance at Regional hospitals by public health COVID-19 response stage.

	Pre-COVID			Containment			Delay			Mitigation			Reopening: Phase One		
Date from	01/01/2020			29/02/2020			13/03/2020			28/03/2020			18/05/2020		
Date to	28/02/2020			12/03/2020			27/03/2020			17/05/2020			31/05/2020		
Number of Days	59			13			15			51			14		
**Daily Visits**			%			%			%			%			%
**Observed Mean**	75		100%	60		100%	46		100%	37		100%	41		100%
**Prior Years Mean**	85		100%	79		100%	85		100%	84		100%	88		100%
**IRR**	0.91			0.75			0.57			0.45			0.50		
**Confidence Interval**	0.86	0.96		0.67	0.83		0.52	0.64		0.42	0.48		0.44	0.56	
***p*-value**	0.001			0.000			0.000			0.000			0.000		
**GP Referrals**															
**Observed Mean**	44		59%	35		59%	17		38%	16		44%	18		44%
**Prior Years Mean**	52		61%	44		51%	49		58%	46		55%	48		55%
**IRR**	0.87			0.77			0.39			0.35			0.40		
**Confidence Interval**	0.79	0.94		0.66	0.91		0.32	0.46		0.31	0.39		0.33	0.48	
***p*-value**	0.001			0.002			0.000			0.000			0.000		
**Admitted**															
**Observed Mean**	20		27%	17		29%	13		28%	11		29%	13		32%
**Prior Years Mean**	20		24%	19		22%	21		25%	19		23%	21		24%
**IRR**	1.04			0.90			0.68			0.56			0.69		
**Confidence Interval**	0.96	1.14		0.77	1.05		0.57	0.80		0.51	0.62		0.58	0.62	
***p*-value**	0.318			0.189			0.000			0.000			0.000		
**Triage urgent (score 1 or 2)**															
**Observed Mean**	20		27%	18		30%	10		23%	8		22%	10		24%
**Prior Years Mean**	21		25%	19		22%	20		23%	21		25%	21		24%
**IRR**	1.01			0.93			0.56			0.39			0.50		
**Confidence Interval**	0.93	1.10		0.79	1.10		0.46	0.67		0.35	0.44		0.41	0.61	
***p*-value**	0.784			0.420			0.000			0.000			0.000		
**Triage low urgency (score 4 or 5)**															
**Observed Mean**	10		13%	6		11%	7		16%	6		16%	6		14%
**Prior Years Mean**	11		13%	11		12%	10		11%	11		13%	13		15%
**IRR**	0.98			0.68			0.76			0.57			0.53		
**Confidence Interval**	0.85	1.13		0.51	0.90		0.59	1.00		0.49	0.67		0.39	0.70	
***p*-value**	0.769			0.008			0.046			0.000			0.000		
**Arrival by ambulance**															
**Observed Mean**	7		9%	7		12%	6		12%	3		8%	3		6%
**Prior Years Mean**	7		8%	7		8%	8		9%	7		8%	8		9%
**IRR**	0.98			1.00			0.78			0.45			0.37		
**Confidence Interval**	0.85	1.12		0.79	1.28		0.60	1.01		0.37	0.55		0.26	0.52	
***p*-value**	0.77			0.98			0.06			0.00			0.00		
**Age category (% proportion)**	*n*	Std Dev		*n*	Std Dev		*n*	Std Dev		*n*	Std Dev		*n*	Std Dev	
**Under 2**	21	6	27%	15	5	24%	15	4	33%	11	3	29%	11	4	27%
**Age 2 to 5**	17	4	22%	14	3	24%	12	4	27%	9	3	25%	8	3	21%
**Age 5 to 12**	24	6	33%	21	5	35%	13	4	29%	12	4	33%	12	3	29%
**Age 13 to 15**	14	5	19%	10	6	17%	6	2	11%	5	2	13%	9	3	23%

*n* is daily unless otherwise stated. 95% confidence intervals; prior year 2018/2019 are averaged mean attendance; percentages are of the total number of visits for the period; while *Phase One* is a three-week period from 18th May to 8th June, this analysis covers the period to 31 May 2020; estimation is by negative binominal regression and IRR is the incident rate ratio; admissions are from ED only.

**Table 4 ijerph-17-06719-t004:** Paediatric emergency department attendance for Complex Chronic Conditions by public health COVID-19 response stage: (Dublin only).

	Pre-COVID			Containment			Delay			Mitigation			Reopening: Phase One			Total		
	2020	2018/ 2019	Change (*n*)	2020	2018/ 2019	Change (*n*)	2020	2018/ 2019	Change (*n*)	2020	2018/ 2019	Change (*n*)	2020	2018/ 2019	Change (*n*)	2020	2018/ 2019	Change (*n*)
**Respiratory**	2	0	2	0	1	−1	0	0	0	1	2	−1	3	0	3	6	3	4
**Endocrine/** **Metabolic**	33	38	−5	9	7	2	3	6	−3	15	24	−9	4	13	−9	64	87	−23
**Digestive**	2	3	−1	0	2	−2	0	1	−1	1	4	−3	0	1	−1	3	10	−7
**Haematology/** **Oncology**	11	23	−12	3	4	−1	1	8	−7	6	16	−10	3	5	−2	24	54	−30
**Total**	48	64	−16	12	13	−1	4	14	−10	23	45	−22	10	18	−8	97	153	−56

Figures are actual number of visits in the period. As 2020 was a leap year, one day extra is included in the 2020 Containment stage; Respiratory includes cystic fibrosis; Endocrine/Metabolic includes diabetes and Addison’s disease; Digestive includes Crohn’s disease, ulcerative colitis and coeliac disease; Haematology/Oncology includes sickle cell anaemia, leukaemia, neoplasm, tumour, space occupying lesion, and other oncology.

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
