# Peer review of "Assessing the Impact of COVID-19 Public Health Stages on Paediatric Emergency Attendance"

_ijerph, 2020, doi:10.3390/ijerph17186719_

Round 1
Reviewer 1 Report
It was a pleasure to review this manuscript regarding decreased Pediatric ED censuses in Ireland during the first three months of the COVID pandemic. In general, there is a great deal of good information here, and I am hoping that the authors can address the concerns I raise below, resulting in an informative, published paper.
My specific comments / suggestions:
Title: I would suggest removing the first four words, or changing the word “dissecting” to something more literal and informative.
Patient population: Although there are suggestions that this paper only studies patients of age 0 to 15, this is not made explicit. It would also be helpful to see some demographic information about the geographic areas being studied.
Also, although the authors assert (lines 59- 60) that this “catchment represents approximately 48% of national annual census of paediatric ED attendances”, this is still unclear – do they mean to suggest that 48% of paed ED attendance comes to the study hospitals, or that 48% of paed ED attendance comes from somewhere in or near Dublin, Limerick, and Cork. In other words, are we likely seeing about half of kids seeking ED care in the country, in this study?
I do not understand why the authors would include an urgent care centre (UCC) in this study of ED attendance. To my understanding, UCCs see lower-acuity patients, and are really different than EDs. Why was this particular one included? Is this the only UCC that sees kids in these cities? Also, if urgent care use has been increasing over the past couple of years, wouldn’t including UCC data distort the historical comparison groups? I note that the “messaging” was specifically worded to include EDs and not UCCs. It also appears, in the footnote to Table 1, that you had to remove UCC data when calculating ICU admissions.
Data: It is unclear how data were acquired. Do the study hospitals use electronic medical records? If so, how were appropriate records identified and data extracted? If not, how were paper charts reviewed? Was there a centralized research database? We need to be confident that we are seeing complete and accurate data.
Line 72: Was the study hospital that stopped seeing children a Dublin or a Regional hospital?
Figure 1 is very nice, and an important component of the paper. I would suggest, though, that the “messaging” be shown in the time periods corresponding to when those messages were circulated, rather than the entire time period. Also, please clarify whether the cases and deaths tallied in “COVID-19” are cumulative or specific to their stages. I would suggest that the latter makes more sense.
Statistical approach: (The line numbers have disappeared from this section, at least on my draft)
Re: “GP referrals” - I am not from your area, so this is mysterious to me. I assume it means patients who were directed to the ED (or maybe UCC) by a general practitioner? What about if the patient was directed to the ED by another physician, such as a paediatrician? Nurse? Others? How were these referrals tracked? Are referral calls or other communications made both to patients / families and to the EDs? In my area, patients who arrive to the ED often tell us that they were referred, but when we attempt verify it (which we usually do not bother doing), often something else happened instead – so, how accurate and complete are your data on this? Needs lots of description, here, since you use these data so much in the analysis.
The “mapping exercise” translating local hospital diagnosis data into ICD data is mysterious, un-described, and un-referenced. This process needs to be explained.
Similarly, the process by which the authors determined “expected prevalence under the restrictive conditions imposed by the pandemic” is also mysterious, un-described, and un-referenced. This process, too, needs to be explained.
The ICD categorization is referenced as [16] but I suspect it is actually [17].
I appreciate the Ethics details.
Results: I would suggest that this section begins with “Pediatric ED…” so it stands alone as a very quotable sentence.
I found this section generally well-done and well-explained.
I would ask the authors to better label the greyed columns in the Tables. Clearly, they are percentages – but of what?
Please define the “Did not wait” category.
Please make clear, in the figure itself, whether the categories in Figure 2 are complaints, diagnoses, something else? How did you manage a patient who would appear in multiple categories?
The first sentence of the second paragraph on page 14 (sorry, still no line numbers) includes the assertion that the Medical category “should be unaffected by the environmental changes imposed by the public health measures” but, again, without methodology, this is just an assertion. Further, I have trouble actually believing this.
How did the authors determine whether or not a patient had a chronic complex condition? This did not seem to be addressed in the Methods section. How accurate was your data source for this determination?
Discussion: The assertion (page 18, line 37) that the higher admission rate of children with behavioral health conditions during the COVID pandemic means that they were more severe, is possibly right, but there are other explanations available. Admissions will also occur when there are fewer barriers, such as when there is bed space for them. An alternative explanation, then, is that barriers were fewer, such as having more bed space was available for a “borderline admit” mental health patient during the COVID outbreak, so more patients were admitted.
The Abstract is generally descriptive of the work.
Author Response
Thank you for taking the time to read our paper and for your feedback. We address each of your suggestions as follows:
Title: I would suggest removing the first four words, or changing the word “dissecting” to something more literal and informative.
The title has now been changed to “Assessing the impact of COVID-19 public health stages on paediatric emergency attendance”.
Patient population: Although there are suggestions that this paper only studies patients of age 0 to 15, this is not made explicit. It would also be helpful to see some demographic information about the geographic areas being studied.
The first paragraph of 2.1 has now been amended to explicitly state the age of patients included in this study and to add some demographic information on the areas supported by the participating EDs and UCC:
“A national database of paediatric emergency department (ED) attendance which would facilitate year on year detailed analysis was not available. With the assistance of on-site system managers, anonymised clinical and demographic data were extracted from the electronic patient records of all children aged under 16 attending three paediatric EDs and an urgent care centre (UCC) in Dublin (referred to as Dublin), and two mixed adult/paediatric EDs located in regional cities (Regional), Limerick and Cork. Approximately 28% of the Irish population live in Dublin, which has 3 dedicated paediatric hospitals and one paediatric UCC. County Cork has a population of about 500,000, representing about 11% of the national population, while County Limerick has a population of approximately 200,000, representing about 4% of the national population. The proportion of the population aged under 16 is comparable in the Cork and Limerick areas, and approximately 2 percentage points higher for the greater Dublin Area [16]. All three Dublin hospitals and the UCC are easily accessed by a large local population, both by car and public transport. However, the population served by the Regional hospitals is more dispersed, with many patients having a longer commute to the ED and more limited public transport options. This is reflected in the proportion of ED visits referred by general practitioners (GPs), with more patients attending a GP prior to visiting the Regional EDs. Medical card status provides us with some information on the socio-economic status of the areas served by each of the participating hospitals. Medical card ownership, which entitles access to health services at no charge and is granted primarily to those on low incomes [17], is above the national average in the areas served by the ED located in Limerick, though below the national average in areas served by the Cork ED. While the Dublin hospitals serve a diverse population and the mix of patients attending each hospital varies in terms of disadvantage, on average, the population in areas served by these hospitals have a medical card ownership rate below the national average [16, 17]. These five hospitals and the UCC represent approximately 48% of national annual census of paediatric (under 16 years) ED attendances [18]. Appendix 1 illustrates the location of each participating site.”
A map identifying the location of the participant hospitals has also been added as Appendix 1.
Also, although the authors assert (lines 59- 60) that this “catchment represents approximately 48% of national annual census of paediatric ED attendances”, this is still unclear – do they mean to suggest that 48% of paed ED attendance comes to the study hospitals, or that 48% of paed ED attendance comes from somewhere in or near Dublin, Limerick, and Cork. In other words, are we likely seeing about half of kids seeking ED care in the country, in this study?
This study captures about 48% of annual paediatric visits nationally. The wording of the final sentence of the first paragraph of 2.1 has been amended to clarify this point:
“These hospitals and the UCC represent approximately 48% of national annual census of paediatric (under 16 years) ED attendances.”
I do not understand why the authors would include an urgent care centre (UCC) in this study of ED attendance. To my understanding, UCCs see lower-acuity patients, and are really different than EDs. Why was this particular one included? Is this the only UCC that sees kids in these cities? Also, if urgent care use has been increasing over the past couple of years, wouldn’t including UCC data distort the historical comparison groups? I note that the “messaging” was specifically worded to include EDs and not UCCs. It also appears, in the footnote to Table 1, that you had to remove UCC data when calculating ICU admissions.
The UCC is located in Dublin and is the only one of its kind in the country. It opened in July 2019 and does, as you note, see low acuity paediatric patients, many of whom might otherwise attend one of the three paediatric EDs included in this study. On the closure of one of the three Dublin paediatric EDs on 28th March, patients were advised to attend either of the other two paediatric EDs or, where appropriate, the UCC. Therefore, excluding the UCC from this analysis would risk overestimating the drop in attendance. That said, the UCC sees far fewer patients than the EDs. For example, 5% of the visits for May included in this study were at the UCC.
The note to Table 1 “Admissions, including ICU admissions, are from ED only” is the clarify that admission figures do not refer to direct admissions, and that as the UCC is not located at a paediatric hospital, it does not admit patients”.
The final paragraph of 2.1 has been reworded as follows to explain why the UCC was included:
“One of the study hospitals in Dublin transferred out all its paediatric services to another location (including redirecting all paediatric ED patients to the two paediatric EDs and the UCC in Dublin) on this date to safely maximise capacity for adult COVID-19 patients at the co-located adult hospital.”
Data: It is unclear how data were acquired. Do the study hospitals use electronic medical records? If so, how were appropriate records identified and data extracted? If not, how were paper charts reviewed? Was there a centralized research database? We need to be confident that we are seeing complete and accurate data.
Electronic records were anonymised on-site by hospital system managers, Paragraph 2.1 (see above) now details this process and hopefully provides assurance on the completeness and accuracy of this data.
Line 72: Was the study hospital that stopped seeing children a Dublin or a Regional hospital?
The study hospital that stopped seeing children was in Dublin and the wording of the final paragraph of 2.1 has been altered to clarify this (see above).
Figure 1 is very nice, and an important component of the paper. I would suggest, though, that the “messaging” be shown in the time periods corresponding to when those messages were circulated, rather than the entire time period. Also, please clarify whether the cases and deaths tallied in “COVID-19” are cumulative or specific to their stages. I would suggest that the latter makes more sense.
Thank you. The figure has been amended to place the messaging under the appropriate stages and the cases and deaths have now been labelled “cumulative”.
Statistical approach: (The line numbers have disappeared from this section, at least on my draft)
Re: “GP referrals” - I am not from your area, so this is mysterious to me. I assume it means patients who were directed to the ED (or maybe UCC) by a general practitioner? What about if the patient was directed to the ED by another physician, such as a paediatrician? Nurse? Others? How were these referrals tracked? Are referral calls or other communications made both to patients / families and to the EDs? In my area, patients who arrive to the ED often tell us that they were referred, but when we attempt verify it (which we usually do not bother doing), often something else happened instead – so, how accurate and complete are your data on this? Needs lots of description, here, since you use these data so much in the analysis.
The Statistical Approach paragraph has now been amended to explain the abbreviation GP:
“Secondary outcomes included referrals by a general practitioner (GP),….”
Patients are rarely referred directly to the ED by a healthcare professional other than their general practitioner. Those who are referred by their general practitioner (GP) are provided with a letter which entitles them to attend the ED free of charge. The usual charge for attending the ED is €100, though is free to those holding a medical card (held by 33% of the total population and provided to those on low income), are entitled to attend the ED free of charge. This fee waiver is in place for GP referrals only, and not for referrals from other healthcare professionals. Paediatricians in Ireland usually work in a hospital setting, and referrals tend to be from the ED to paediatrics, often on an out-patient basis.
The final paragraph of the discussion (before section 4.1) has been amended to clarify this:
“The decline in the proportion of GP referred visits may be due to an overall decrease in attendance by paediatric patients at general practice. Many GPs moved to remote consultations from the onset of the pandemic, conducting fewer face-to-face consultations. GPs also played a critical role in referring probable cases of COVID-19 for testing, so accessibility for non-COVID illness may have been an issue. However, the decreased GP referral rate may also be somewhat attributable to a change in the referral process. GP referrals are captured on ED systems on the presentation of a referral letter and, as a GP referral entitles patients to a waiver of the usual ED charge of €100 for those without a medical card, capture of such referrals on ED systems is generally robust. However, some remote consultations may have resulted in advice to attend the ED without a GP issuing an accompanying letter. “
The “mapping exercise” translating local hospital diagnosis data into ICD data is mysterious, un-described, and un-referenced. This process needs to be explained.
Appendix 3 has been added which details the conditions mapped under each of the more granular categories in Figure 2 and the summary of these categories as per Figure 3.
Similarly, the process by which the authors determined “expected prevalence under the restrictive conditions imposed by the pandemic” is also mysterious, un-described, and un-referenced. This process, too, needs to be explained.
This has been clarified by the addition of examples as follows to text to section 2.2:
“ED diagnosis data was available for the Dublin hospitals, however, as none of these hospitals used matching diagnostic coding, a detailed mapping exercise was conducted, using International Classification of Disease (ICD) categorisation as a guide [19] and with input from ED and paediatric clinicians, to ensure consistent categorisation across hospitals (Appendix 3). Categorisation was further consolidated into summarised category of conditions based on expected prevalence under the restrictive conditions imposed by the pandemic related public health responses. For example, all conditions predominantly contagious in nature were likely to be impacted in a similar manner by the low level of personal contact (“Infectious”), while the suspension of sporting and other physical activities, coupled with the requirement to remain close to home, were likely to have a similar effect on many types of injuries (“Trauma and Physical Activity“). Graphic analysis illustrates weekly attendance by summary category and variation on prior year by public health stage for the more detailed categories.”
The ICD categorization is referenced as [16] but I suspect it is actually [17].
This has now been corrected. Thank you.
I appreciate the Ethics details.
Thank you.
Results: I would suggest that this section begins with “Pediatric ED…” so it stands alone as a very quotable sentence.
Amended.
I found this section generally well-done and well-explained.
Thank you.
I would ask the authors to better label the greyed columns in the Tables. Clearly, they are percentages – but of what?
The note to Table 1 has been amended to clarify this:
“Percentages are of the total number of visits for the period.”
Please define the “Did not wait” category.
The note to Figure 3 now includes the following:
”Did Not Wait refers to patients who registered on arrival but did not wait to see a doctor. This varies with the average wait time and is generally higher during busy period.”
Please make clear, in the figure itself, whether the categories in Figure 2 are complaints, diagnoses, something else? How did you manage a patient who would appear in multiple categories?
The title of Figure 2 has been amended to “Change by Diagnosis from prior year by Public Health Stage”.
The paragraph describing Figure 2 now includes an explanation of categorisation for patients with multiple diagnoses:
“Figure 2 charts the percentage change compared to prior years for each of the main categories of diagnosis for the Dublin hospitals by public health stage, with a detailed breakdown reported in the Appendix 2. For patients with multiple diagnoses, the first diagnosis is used in this analysis.”
The first sentence of the second paragraph on page 14 (sorry, still no line numbers) includes the assertion that the Medical category “should be unaffected by the environmental changes imposed by the public health measures” but, again, without methodology, this is just an assertion. Further, I have trouble actually believing this.
Appendix 2 details of conditions defined within the category “Medical”. This category includes conditions such as appendicitis, urinary tract infections, cellulitis, diabetes mellitus. Most attendances falling within this category are not contagious or related to activities such as sports, and therefore attendances for these reasons might be expected to be relatively stable. However, attendance for some of the conditions within this category could be delayed due to apprehension/fear on behalf of the parents due to COVID-19.
The wording has been altered as follows:
“The summary category Medical, which groups clinical presentations that may be unaffected by the environmental changes imposed by the public health measures.”
How did the authors determine whether or not a patient had a chronic complex condition? This did not seem to be addressed in the Methods section. How accurate was your data source for this determination?
Section 2.2 of the Methods section now includes a definition of chronic complex condition, including a supporting reference, and Appendix 3 details the conditions included in this category.
Discussion: The assertion (page 18, line 37) that the higher admission rate of children with behavioral health conditions during the COVID pandemic means that they were more severe, is possibly right, but there are other explanations available. Admissions will also occur when there are fewer barriers, such as when there is bed space for them. An alternative explanation, then, is that barriers were fewer, such as having more bed space was available for a “borderline admit” mental health patient during the COVID outbreak, so more patients were admitted.
That is a very valid suggestion and the wording of this section has been amended to incorporate this possibility:
“However, as the admission rate of these presentations is higher than prior years, potentially indicating a greater severity of presentation, this may be reflective of delayed presentation. The lack of community mental health services and supports due to the constraints of the public health measures may also explain this increased severity. Increased admissions may, however, indicate greater capacity to admit, particularly cases that may be perceived as “borderline admit”.
The Abstract is generally descriptive of the work.
Thank you.
Reviewer 2 Report
In this study, the authors investigate the impact of COVID-19 on the use of paediatric emergency departments in Ireland. The article is well written and deals with a current topic that is of interest for publication. My only comment is whether it would be possible to expand figure 3 further from May 26th as the effect of the reopening phase cannot be assessed in such a few days.
Author Response
.My only comment is whether it would be possible to expand figure 3 further from May 26th as the effect of the reopening phase cannot be assessed in such a few days.
Thank you for taking the time to read this study. Figure 3 has now been updated to the end of June 2020.
Reviewer 3 Report
This is a very interesting and worth reading article. Still, I recommend to further improve the following items:
- It would be interesting to have a map with the location of the hospital facilities under study;
- A socioeconomic characterization of the different regions at stake should be provided, which should be analysed to see if they can support/explain some of the results attained;
- The discussion section could be improved in what regards the accessibility issue (e.g. can the restrictions to public transports partly explain a reduction of attendance?). There are tons of literature available on accessibility to healthcare facilities;
- Point 2 of 4.2.1 key message could be more detailed in the introduction section. This could provide more robustness to the article.
Author Response
Thank you for taking the time to review our study and for your feedback. We address each of your concerns below:
- It would be interesting to have a map with the location of the hospital facilities under study;
A map of the location of the six participating sites is now included at Appendix 1.
- A socioeconomic characterization of the different regions at stake should be provided, which should be analysed to see if they can support/explain some of the results attained;
Further information on the characteristics of the regions included in this study has been added to 2.1 Data as follows:
“A national database of paediatric emergency department (ED) attendance which would facilitate year on year detailed analysis was not available. With the assistance of on-site system managers, anonymised clinical and demographic data were extracted from the electronic patient records of all children aged under 16 attending three paediatric EDs and an urgent care centre (UCC) in Dublin (referred to as Dublin), and two mixed adult/paediatric EDs located in regional cities (Regional), Limerick and Cork. Approximately 28% of the Irish population live in Dublin, which has 3 dedicated paediatric hospitals and one paediatric UCC. County Cork has a population of about 500,000, representing about 11% of the national population, while County Limerick has a population of approximately 200,000, representing about 4% of the national population. The proportion of the population aged under 16 is comparable in the Cork and Limerick areas, and approximately 2 percentage points higher for the greater Dublin Area [16]. All three Dublin hospitals and the UCC are easily accessed by a large local population, both by car and public transport. However, the population served by the Regional hospitals is more dispersed, with many patients having a longer commute to the ED and more limited public transport options. This is reflected in the proportion of ED visits referred by general practitioners (GPs), with more patients attending a GP prior to visiting the Regional EDs. Medical card status provides us with some information on the socio-economic status of the areas served by each of the participating hospitals. Medical card ownership, which entitles access to health services at no charge and is granted primarily to those on low incomes [17], is above the national average in the areas served by the ED located in Limerick, though below the national average in areas served by the Cork ED. While the Dublin hospitals serve a diverse population and the mix of patients attending each hospital varies in terms of disadvantage, on average, the population in areas served by these hospitals have a medical card ownership rate below the national average [16, 17]. These five hospitals and the UCC represent approximately 48% of national annual census of paediatric (under 16 years) ED attendances [18]. Appendix 1 illustrates the location of each participating site.”
- The discussion section could be improved in what regards the accessibility issue (e.g. can the restrictions to public transports partly explain a reduction of attendance?).
The discussion has been amended, including the addition of three supporting references, as follows:
“Furthermore, as availability of transport to and from EDs is a concern for many parents [21, 22], public health restrictions on travel and fear of using public transport may have led to some parents deciding not to access needed healthcare [23].”
Section 2.1 now also includes the following:
“All three Dublin hospitals and the UCC are easily accessed by a large local population, both by car and public transport. However, the population served by the Regional hospitals is more dispersed, with many patients having a longer commute to the ED and more limited public transport options. This is reflected in the proportion of ED visits referred by general practitioners (GPs), with more patients attending a GP prior to visiting the Regional EDs”.
- Point 2 of 4.2.1 key message could be more detailed in the introduction section. This could provide more robustness to the article.
This section has been reformatted as the Conclusion and additional text has been added as advised:
“6. Conclusions
Presentations to the ED have declined sharply, particularly among children, in the weeks immediately following the onset of the COVID-19 pandemic. Delayed or reduced access to paediatric emergency care can have severe consequences and concern has been expressed that the reduction in ED attendance may be due to avoidance behaviour. This study uses a unique sample of paediatric ED attendance to understand changes in ED presentations from the most restrictive phase of public health COVID-19 response through to the beginning of reopening. While a reduction in attendance for respiratory conditions, infectious diseases and certain injuries can be attributed to the restrictive living conditions imposed in response to COVID-19, a sustained decrease in medical attendance, and for complex chronic conditions in particular, may be indicative of avoidance behaviour. Decreased referrals from GPs may be indicative of reduced accessibility to general practice due to the pressure on GPs to manage the response to COVID-19. Surveillance of the nature of paediatric ED attendance must continue to ensure the identification of those at risk due to delayed attendance and to inform public health messaging as the burden placed on families of living through a pandemic persists.”
Round 2
Reviewer 1 Report
thank you for addressing my concerns
Reviewer 3 Report
The authors did an excellent job in trying to include most of the comments previously made. In my opinion, the article can be published as it is.